# A Simulation Study on the Processes of Intra-Group Informal Interaction Affecting Workers’ Safety Behaviors

**DOI:** 10.3390/ijerph191610048

**Published:** 2022-08-15

**Authors:** Huihua Chen, Cong Chen, Hujun Li, Jianshe Zhang, Zengke Yang

**Affiliations:** 1Department of Engineering Management, School of Civil Engineering, Central South University, Changsha 410075, China; 2School of Civil Engineering, Henan Polytechnic University, Jiaozuo 454003, China

**Keywords:** intra-group informal interaction, workers’ safety behaviors, group knowledge sharing, group identification, simulation analysis

## Abstract

The construction industry across the world is characterized by a high safety risk, and the occurrence of these safety accidents has led to substantial economic and social losses. The workers’ unsafe behaviors are considered to be a main cause. Thus, recently, scholars in the construction industry have shifted their attention to the investigation of the influencing factors (or antecedents) and their impact on workers’ safety behaviors (WSBs), hoping to provide insight into useful management policies. The existing literature has identified many society-level, cooperation-level, project-level, and individual-level concepts influencing WSB, but ignores the influence of intra-group informal interaction (IGII) on WSB. This study constructed a conceptual model for IGII, group knowledge sharing (GKS), and group identification (GI) to determine their influence on construction workers’ safety behaviors, and then conducted simulation analysis using the software of NetLogo. The results show that IGII, GKS, and GI can positively influence workers’ safety behaviors, and IGII can also positively influence WSB through GKS and GI. This study enriches the theoretical knowledge on the causation of construction workers’ safety behaviors, provides references for project managers to carry out proper safety management, and offers a theoretic foundation for the formulation of industry regulations.

## 1. Introduction

The construction industry is characterized by higher safety risks, when compared to other industrial sectors (such as manufacturing) [1,2]. Numerous construction safety accidents have happened all over the world, resulting in substantial monetary and non-monetary losses for the nation, the society, the construction sector, the business, and more seriously for the construction workers themselves [3,4]. Therefore, investigating the causation of construction safety accidents and developing appropriate management methods to lessen or even prevent these mishaps have attracted great academic interest [5]. Construction workers’ safety behaviors (WSBs) are widely known as workers’ positive or negative actions related to construction safety performance. When examining the causation of construction safety accidents in detail, many academics have noted that the WSB can be directly linked to the safety accident, highlighting that workers’ unsafe behaviors are the primary culprit [6,7,8]. The solution to construction safety issues has thus gradually focused on the development of scientific procedures that encourage and motivate construction workers to adopt safety behaviors.

The questions triggered many scholars’ interests, and the existing literature has identified numerous antecedents of WSB. These antecedents can be categorized into five groups, i.e., individual characteristics, group interactions, work and workplace design, project management and organization, and family, industry, and society [6]. It was discovered that these previous studies had investigated WSB based on the project governance structure established on formal rules, agreements, and employment contracts, while ignoring the fact that China’s construction workers’ behavioral decision making is influenced by their indigenous settings. The employment reform based on contract theory in the last two decades has not greatly altered the conventional employment pattern in China’s construction industry. China’s construction laborers typically belong to an indigenous working group, with small groups serving as the fundamental work unit (including about seven to eight workers). The small working group consists of a foreman (the actual owner or manager of the group, called “baogongtou” in Chinese) and some workers who are related to the foreman through kinship, marriage, or townsman connections [9,10]. Unlike small independent contractors serving in the Western construction sector, these small working groups in China are not registered formal companies, and the workers in these groups rarely have formal employment contracts with the foreman (or there is no need for a written contract). The small working groups are built and maintained by their members’ informal interactions based on these aforementioned ties (i.e., kinship, marriage, and townsmen ties), namely, intra-group informal interaction (IGII).

Because of their extensive life experience, China’s construction workers believe that the aforementioned informal interactions might foster trust or provide advantages as opposed to formal interactions based on formal agreements and employment contracts. As a result, every construction worker should maintain informal communication and contact with the foreman and his/her coworkers. These informal interactions are critical factors when making behavioral decisions. Despite the literature, which reveals a wide range of influential factors affecting construction WSB, there exist limited studies on IGII, and its effect on WSB. Group knowledge sharing (GKS) is referred to the process of transferring and sharing knowledge in the group through various channels, in which group members absorb knowledge and apply it to the best of their ability [11,12]. During construction workers’ informal interactions, work-related information or knowledge will be transferred and shared among them. Thus, GKS might exert an effect on the relationship between IGII and WSB, and the previous literature cannot give detailed answers. Additionally, group identification (GI) is the degree of workers’ recognition and acceptance of the group to which they belong. Workers may more easily build mutual identification when informal interactions occur frequently, which can strengthen GI. Thus, we argue GI might also have an effect on the relationship between IGII and WSB. Yet, there exist few studies that examine the inference.

The purpose of this study was to look into the issues raised above. We first established a conceptual model to show the relationships among IGII, GKS, GI, and WSB. We then designed a simulation model using the software of Netlogo to test the relationships among these related concepts. The results of the study can provide a reference for managers in taking reasonable actions to encourage WSB, assist project managers in developing effective safety management policies, and serve as a foundation for regulators in establishing industry rules.

## 2. Literature Review and Conceptual Model

### 2.1. Intra-Group Informal Interaction

Informal interaction can be referred to every two individuals’ linguistic and behavioral interactions based on informal relationships (e.g., a kinship relationship), which are established based on particularism [13]. The main differences between informal interactions and formal interactions are that: (a) formal interactions can occur because of the formal relationship between both parties, and these relationships are stated in the employment contracts or agreements; (b) the objective of these interactions is mostly work tasks and feedback. Related studies have pointed out that informal interaction is a prevalent phenomenon in the political organizational context [14,15], social organizational context [16,17], and business/business organizational settings [18].

Numerous research has suggested that informal interaction is a concept that exists at the individual level [19]. The concept denotes a dual interaction between two parties that is developed and maintained to promote human exchanges and is based on particularism and emotion [20]. Some other researchers argued that informal interaction can occur at the organizational level. They viewed organizational informal interaction as social capital and a strategic tool that aids in organizational functioning, offers opportunities for dialogue, helps gather information, and assists in establishing trust [21,22,23]. According to Luo and colleagues [24], informal interaction refers to a range of interpersonally oriented actions that organizational managers engage in with their business partners.

Informal interaction is prevalent in China’s society mainly because it is a universal mechanism for allocating or acquiring resources (similar to the bureaucratic system and market processes) [25,26]. China’s construction workers are typically part of indigenous groups, which are mostly formed and maintained through kinship relationships, marital relationships, and fellow-township relationships [10]. IGII for them can be referred to as informal communication, exchanges, and transactions between workers and coworkers and workers and foremen. In this study, we defined construction workers’ IGII as the establishment or maintenance of personal bonds among workers during the course of construction work that involve verbal, behavioral, and emotional exchanges between two parties. The typical IGII behaviors of Chinese construction workers are listed in Appendix B.

### 2.2. Group Knowledge Sharing

Group knowledge sharing (GKS) can be traced back to organizational knowledge sharing. Knowledge sharing refers to the transfer and exchange of knowledge between individuals, groups, and organizations through a variety of sharing methods [27,28]. It has a considerable impact on how organizational members behave [29]. Work-related information within an organization or group provides a powerful competitive advantage; thus, these entities will act sensibly to secure the acquisition, storage, and transfer of pertinent knowledge [30].

Knowledge sharing can have a direct influence on organizational members’ innovative behavior [31,32,33], participation behavior [34], organizational citizenship behavior [35,36], trust behavior [37], and safety behaviors [38,39]. For instance, Ramasamy et al. [35] studied the relationship between knowledge sharing and organizational citizenship behavior and argued that all five components of organizational citizenship behavior are positively related to knowledge sharing; Lee et al. [39] reported that knowledge sharing can directly influence employee’s safety behavior. Construction workers in China typically share relevant information in the working group. In this paper, we define GKS as the behavior of exchanging and sharing knowledge, including experience, technology, and safety information, among group members in order to achieve construction goals.

### 2.3. Group Identification

Group identification (GI) is derived from organization identification, which was established based on social identity theory and self-classification theory [40]. Previous scholars pointed out that in an organization built on work groups, the individual’s identification with the group is more effective than his/her identification with the organization in explaining the individual’s attitude and behavior [41]; as a result, GI became the focus of management researchers.

GI is referred to the degree of internal members’ recognition and acceptance of the group to which they belong [40,42]. The concept denotes an individual’s emotional tendency to self-identify as a group member. This emotional propensity stands for the individual’s recognition and acceptance of group members into the group to which they belong. Previous studies reported that GI can be divided into three dimensions, i.e., cognitive, evaluative, and affective [40], and confirmed that GI is a reliable indicator of individual behavior, individual competence, and psychological traits [43,44,45], as well as group performance [46,47,48]. As for construction workers, GI is the extent to which the group’s perception is acknowledged and accepted through techniques such as affective connection and self-categorization.

### 2.4. Workers’ Safety Behaviors

The academic research on WSB can be traced back to the accident causation theory. According to Heinrich’s domino theory, safety accidents are primarily caused by workers’ unsafe behaviors [49]. However, the concept of WSB was not initially embraced by academia. More researchers are turning to WSB as the notion of behavior-based safety gains traction. They emphasize that WSB can be a more accurate predictor of safety performance when compared to accident fatality and injury rates, and that looking into the incentive mechanisms of safety behaviors can provide some ex ante strategies to reduce safety accidents [50].

WSB is not clearly defined by previous scholars, and the basic rule is that the concept or term is “seeing then knowing”. Broadly speaking, WSB can be described as some sets of actions related to direct safety performance (i.e., fatality or injury); thus, workers’ safety behaviors (positive behavior that can reduce fatality or injury), workers’ unsafe behavior (negative behavior that can cause fatality or injury), and safety citizenship behaviors all encompass WSB. However, in a limited sense, WSB is only connected to a few worker safety behaviors (such as wearing safety helmets and safety belts) [51]. This definition is also widely accepted by area researchers.

WSB might be viewed as a one-dimensional concept or a multi-dimensional concept in the extant literature. The first point of view measures the concept as a whole, while the second view argues that WSB should include some components or dimensions. The widely accepted interpretation is that WSB can be divided into safety compliance behavior and safety participation behavior [52,53]. Safety compliance behaviors involve adhering to safety procedures and carrying out work in a safe manner, and safety participation behaviors involve helping coworkers, promoting the safety program within the workplace, demonstrating initiative, and putting effort into improving safety in the workplace [54]. Other scholars offered some different views. For instance, Andriessen [55] divided WSB into attentive behavior and safety active behavior; Larsson et al. [56,57] classified WSB into structural safety behaviors, interactive safety behaviors, and personal safety behaviors; and Gao et al. [58] classified WSB into task performance safety behaviors and situation performance safety behaviors. In this study, we selected the highly cited dimensions of WSB, i.e., safety compliance behaviors and safety participation behaviors. The detailed actions checklist is provided in Appendix C.

### 2.5. IGII and WSB

IGII involves a worker’s informal interactions with his leader and his colleagues or workmates in the working group. Few scholars have investigated the relationships between IGII and WSB, but the existing literature on formal interactions can provide some information for postulation. Previous research has identified that a worker’s formal interactions with his leader can influence his safety behaviors. For instance, He et al. [59] reported that high-quality leader–member exchange within a group can enhance WSB; Burns et al. [60] highlighted that safety transformational leadership can directly influence WSB and indirectly influence WSB through intrinsic motivation. Additionally, a worker’s colleagues also can affect his (her) safety behaviors. For example, Kaufman et al. [61] clarified that safety support from group leaders can motivate workers’ safety behaviors; Liang et al. [62] analyzed the social contagion effect of coworkers’ safety violations of WSB within a group; Choudhry et al. [63] argued that some unsafe behaviors of workers are due to coworker pressure; and Shi et al. [64] clarified that workmates can influence WSB through their positive reinforcing behaviors, and influence workers’ unsafe behaviors through negative reinforcing behaviors.

As previously highlighted, China’s construction worker groups are typical groups based on informal relationships. Formal contracts and agreements are documents only utilized to deal with inspections by top managers or government officials, and the transactions between two parties are actually bound and governed by behaviors based on informal relationships, namely IGII. Firstly, IGII in Chinese construction working groups plays the same role as the formal interactions in other formal working groups. Secondly, Chinese construction workers’ IGII is based on informal relationships, and it can facilitate mutual recognition and understanding among workers, and accordingly can lead to behaviors expected by the working group or behavioral consistency with coworkers. Hence, we presume that construction workers’ IGII can influence WSB.

### 2.6. The Impact of GKS

For China’s construction workers, there exist many different kinds of informal interactions in the working group. These informal interactions involve information, expertise, and resource (materials, equipment, etc.) exchanges [65,66]. IGII is hence an important way to maintain GKS. In addition, safety education and training in China largely focuses on managers, whereas few front-line workers are included. Construction workers’ safety knowledge is mostly acquired through informal interactions with their foreman and coworkers. For instance, the foreman and elder workers are more likely to share their safety knowledge and experience when drinking and chit-chatting after work. As previously established (see Section 2.2), GKS is a significant influencing factor of worker’s behavior, as workers can acquire the prerequisite knowledge for behavioral decision making through GKS. Additionally, GKS is also an influential factor in project performance [67,68]. For instance, Jian et al. [69] and Zhu et al. [70] reported that knowledge sharing during construction projects has a positive impact on project performance; Jamshed [71] also highlighted that knowledge sharing has a significant effect on team performance. However, project performance can be achieved only when front-line workers are conducting group-expected behaviors. Thus, IGII can influence GKS, GKS can influence WSB, and IGII can also influence WSB through GKS.

### 2.7. The Impact of GI

Previous research suggested that workers’ informal interaction in the group might be an influencing variable of GI. Firstly, IGII involves mutual emotional communication between group members, and better IGII can lead to mutual understanding and collaboration among group members, which can strengthen the workers’ identification with the group [72]. Secondly, better IGII indicates better relationship quality between workers, which can make employees feel “at home” in the group. This kind of feeling reflects the workers’ identification with the group [72].

When group members have a high level of GI, they might be more likely to adopt group-expected actions. The existing literature reported that GI can promote workers’ hard-working behavior [47], innovative behavior [73,74], mutual helping behavior [75,76], and organizational citizenship behavior [77]. The reasons for this are that: (a) when a group member has a strong sense of identification with his/her working group, he/she might identify himself/herself with the team as a community of interest; the achievement of his/her goals is based on or included in the process of achieving group goals; and he/she hence will tend to adopt behaviors the group expects [47] and (b) a higher GI typically denotes stronger emotional connections among group members, as a worker might take into account other workers’ concerns when behaving; thus, certain actions (such as mutual helping behavior) might be carried out in a specific setting to serve the other workers’ interests [77,78]. This logic might be followed when workers conduct safety behavior decision making. Therefore, we argue that IGII can influence GI, GI can influence WSB, and IGII can influence WSB through GI.

Based on above analyses, we constructed a conceptual model of this study (see Figure 1), which was then used as the basis of a simulation model.

## 3. Simulation Experiment Design

The method used for the computational experiment was employed to carry out simulations of this study. We selected NetLogo 6.2.0 to realize the simulation. NetLogo is a multi-agent simulation platform which is simple and easy to operate. This software provides a large scale of demo cases, which allow the users to simulate natural and social phenomena. The NetLogo platform can realize agent-based modeling and system dynamics modeling, and it is also widely used for simulation of individual and group behavior in the social sciences. Therefore, the NetLogo platform is suitable for simulating the influence of IGII on WSB.

### 3.1. Simulation Flow

The default zoom of NetLogo was utilized as a virtual construction site to display the simulation. The agents on the site were Patches and Turtles. Thus, we reconstructed the Patch class to denote the external context, and re-established the Turtle class to denote the front-line construction worker. Turtles (workers) interacted with Patches (external context) during construction.

According to the theory of planned behavior, the workers’ willingness to conduct safety behavior is the key to WSB. As many factors are reported to have influences on WSB willingness (e.g., safety attitude, safety climate, social capital) [51], we employ the term external context to generally refer to the group-level, project-level, industry-level, and society-level factors, and utilize the term of internal setting to refer to the individual factors in this study. Thus, IGII, GKS, and GI, as highlighted in this study, are also included in the external context. Workers’ safety behavior cost (WSBC), being interpreted as the physical and mental barriers to successful WSB, can be viewed as a salient individual-level factor that reduces WSB willingness. During the agents’ random interaction, if workers were impacted by external contextual factors and internal setting factors, they would demonstrate either enhanced and reduced willingness to engage in safety behaviors; and if they were not affected, then they would enter the next round of random interaction. Then, if workers’ willingness to behave safely gradually increased with the interaction, these workers would be likely to adopt safety behaviors when they showed positive willingness, while if workers showed negative willingness to behave safely, they entered the next round of random interaction, as shown in Figure 2.

According to Figure 2, the specific steps of the simulation were as follows:

Step 1. Utilize the default zoom of NetLogo to build the virtual construction site.

Step 2. Reconstruct the Patch class and Turtle class to represent construction external context and construction workers. The added properties of agents can be seen in Table 1.

Step 3. Design the action rules of the agents. We set three action rules, including a Working rule, Being_cultivated rule, and Producing_SB rule. The Working rule described the workers’ working track; the Being_cultivated rule defined the influencing degree of external context factors on WSB willingness, which was calculated based on a questionnaire (see Appendix A for relevant information); the Producing_SB rule defined the probability of workers adopting safety behaviors when WSB willingness was positive (see in Table 1).

Step 4. Realize the simulation model on the Netlogo. This step included compiling the back-end codes and designing the front-end interface.

Step 5. Design the experimental scenarios and conduct experimental simulations.

### 3.2. Variable Parameter Setting

The variable parameters of the simulation model were set, and the variables mainly included external context variables and agent-related variables. The variable parameters are detailed in Table 1.

The IGII, GKS, and GI properties of Patches: the initial values of the properties of each patch were random, and they were set to satisfy a normal distribution with a mean of the initial values and a variance of 1.

WSBC and WSB willingness caused by other factors: every worker’s safety behavior cost and safety behavior willingness were different; they were set to satisfy a normal distribution with a variance of 1.

Safety behavior generation coefficient: when WSB willingness was positive, the worker had a high probability of adopting safety behaviors; hence, the safety behaviors formation coefficient was set to satisfy the binomial distribution of *p* = 0.8.

## 4. Simulation Experiments

### 4.1. Single Variable Experiment

#### 4.1.1. Simulation Experiment for Different IGII Scenarios

IGII was used as the adjustable variable, and other variables were set as initial values. The low-level scenario (IGII = 2) and high-level scenario (IGII = 8) were set, and the variation in WSB under the two scenarios was obtained by simulation analysis, as shown in Figure 3.

When IGII = 2, all the variables were set to initial values and the workers’ perceived IGII was at a low level. When t = 0, all construction workers showed a low willingness to behave safely. Starting from t = 15, some construction workers showed high willingness to engage in safety behaviors and started to adopt safety behaviors; and after t = 24, the number of construction workers who adopted safety behaviors remained relatively stable.

When IGII = 8, the other variables were initial values and the workers’ perceived IGII was at a high level. The overall growth with time of the construction workers’ safety behaviors was roughly the same as that when IGII = 2. However, when t = 12, workers started to adopt safety behaviors; when t = 18, the number of workers who adopted safety behaviors reached a relatively high point, and subsequently fluctuated around this value.

The comparative analysis found that when IGII increased, after a shorter period, almost all workers chose to adopt safety behaviors, which shows that a high level of IGII will help enhance their willingness to act safely and thus adopt safety behaviors.

#### 4.1.2. Simulation Experiment for Different GKS Scenarios

GKS was used as the adjustable variable and other variables were set as initial values. The low-level scenario (GKS = 2) and high-level scenario (GKS = 8) were set, and the variation in WSB under the two scenarios was obtained by simulation analysis, as shown in Figure 4.

When GKS = 2, all the variables were set to initial values and the workers’ perceived GKS was at a low level. When t = 0, all construction workers showed a low willingness to behave safely. Starting from t = 9, some construction workers showed high willingness to engage in safety behaviors and started to adopt safety behaviors; and after t = 11, the number of construction workers who adopted safety behaviors remained relatively stable.

When GKS = 8, the other variables were initial values and the workers’ perceived GKS was at a high level. The overall growth with time of the construction workers’ safety behaviors was roughly the same as that when GKS = 2. However, when t = 7, workers started to adopt safety behaviors; when t = 10, the number of workers who adopted safety behaviors reached a relatively high point, and subsequently fluctuated around this value.

The comparative analysis found that when GKS increased, after a shorter period, almost all workers chose to adopt safety behaviors, which shows that a high level of GKS will help enhance their willingness to act safely and thus adopt safety behaviors.

#### 4.1.3. Simulation Experiment for Different GI Scenarios

GI was used as an adjustable variable, and other variables were set as initial values. Two scenarios, i.e., low safety behavior cost (GI = 2) and high cost of safety behaviors (GI = 8), were set to conduct the simulation, and the changes in WSB under the two scenarios are shown in Figure 5.

When GI = 2, all the variables were set to initial values and the workers’ perceived GI was at a low level. When t = 0, all construction workers showed a low willingness to behave safely. Starting from t = 10, some construction workers showed high willingness to engage in safety behaviors and started to adopt safety behaviors; and after t = 14, the number of construction workers who adopted safety behaviors remained relatively stable.

When GI = 8, the other variables were initial values and the workers’ perceived GKS was at a high level. The overall growth with time of the construction workers’ safety behaviors was roughly the same as that when GI = 2. However, when t = 7, workers started to adopt safety behaviors; when t = 11, the number of workers who adopted safety behaviors reached a relatively high point, and subsequently fluctuated around this value.

With IGII, GI, and GKS being set as adjustment variables and other variables being set as initial values, four scenarios were set: (IGII, GI, GKS) = (2, 2, 2), (IGII, GI, GKS) = (8, 2, 2), (IGII, GI, GKS) = (8, 8, 2), and (IGII, GI, GKS) = (8, 8, 8). The variation in WSB for all the scenarios was obtained by simulation analysis, as shown in Figure 6.

When (IGII, GI, GKS) = (8, 8, 2), the other variables were set as initial values and both workers’ IGII and GI were at high levels. In this scenario, the overall rising tendency of construction workers to engage in safety behaviors was roughly the same as that when IGII = 2. When t = 7, construction workers started to adopt safety behaviors; and from t = 10, the workers who adopted safety behaviors reached a relatively high point with subsequent random fluctuations around that value, while in the scenario of (IGII, GI, GKS) = (8, 8, 8), construction workers started to adopt safety behaviors at t = 6, and from t = 8, the workers who adopted safety behaviors reached a relatively high point with subsequent random fluctuations around that value.

Based on the comparative analyses, it can be found that when IGII, GI, and GKS all increased, most workers adopted safety behaviors after a shorter period of time, which shows that the combined effect of high levels of IGII, GI, and GKS was more likely to enhance workers’ willingness safety behavior and thus, adopt safety behaviors.

## 5. Discussion

### 5.1. Impact of IGII on WSB

The study results show that IGII can positively affect WSB, which can be explained based on the existing literature. As previously established, IGII involves a worker’s interactions with his/her foreman and his/her coworkers. Construction foremen or supervisors play a significant role in changing WSB [79,80]. Firstly, the front-line supervisors can conduct safety training and preventive action, reactive and supportive action, and safety communication on the construction site [81,82]. These supervisors’ behaviors can not only directly influence WSB, but can indirectly impact WSB through creating a safety climate and psychological contract [81,82,83]. Secondly, supervisors’ leadership practices include transformational and contingent reward leadership, two types of leadership associated with greater levels of safety compliance and safety participation [84]. In addition, coworkers can also exert significant effects on WSB. The reason for this is the existence of peer pressure [63] and social contagion effects [62,85]. When their co-workers showed some risky behaviors, workers were inclined to learn these behaviors, because these behaviors make work more convenient and energy-saving, or show masculinity [86].

### 5.2. The Role of GKS and GI

The simulation showed that both GKS and GI can positively influence IGII’s effect on safety behaviors, which can also be explained based on the existing literature. Research on intra-organizational interactions pointed out that intra-organizational interactions are prerequisites for intra-organizational knowledge sharing, and can facilitate organizational knowledge transfer and sharing [87,88]. Additionally, organizational knowledge sharing was reported as a significant influencing factor of members’ behaviors within an organization, and was the basis of behavioral decisions [31,34]. Research on intra-organizational interactions also reported that intra-organizational interactions can facilitate mutual emotional interactions, which can lead to the members’ identification with the organization. In addition, related studies have shown that GI originates from organizational identification, which directly affects organizational constructive behaviors [48,76], mutual helping behaviors, and organizational behaviors [77].

### 5.3. Management Insights

#### 5.3.1. Stimulate the Combined Effect of IGII

China’s construction working groups are established and maintained through IGII, and IGII is a stronger social constraint for every construction worker. As reported in the simulation experiment, in the scenario of higher IGII, GKS, and GI, construction workers are more willing to adopt safety behaviors. These three concepts together exert a salient effect on WSB. Therefore, group foremen should first pay attention to the role of IGII in constructing GI and promoting GKS, and use communication and other methods to strengthen workers’ group identification and promote communication. At the same time, other methods should be designed to cultivate GI and GKS, and IGII can better exert positive effects on WSB when workers have a stronger identification with the group or foreman and a higher quality of GKS.

#### 5.3.2. Using the Positive Effects of GKS and GI

GKS involves group members’ learning and accumulation of safety knowledge, which are the basis for safety behavior decision making. Group supervisors or foremen can issue management instructions more efficiently, and group aims can be better achieved when group members identify with their working group. GI is derived from the theory of social identity, and an individual’s self-cognition process is the precondition for social identification. The cognitive process of group members can be effectively promoted through the establishment of the GI climate. Additionally, the GKS climate is a significant influencing factor for knowledge sharing in groups. For a working group, if there is no harmonious knowledge sharing climate, group members are more likely to reduce their willingness to share knowledge to protect themselves.

## 6. Conclusions

### 6.1. Key Findings

(1).Based on the existing literature, a conceptual model of IGII’s effects on WSB was established. The conceptual model involved IGII, GKS, GI, and WSG, indicating the relationship between these concepts.(2).A simulation model was constructed based on the conceptual model using a computational experiment. Then, we conducted single-variable experiments and multi-variable experiments to demonstrate the change in WSB under the influence of IGII, GKS, and GI. These experiments validated that IGII, GKS, and GI can all positively influence WSB; IGII can also positively influence WSB through GKS and GI.

### 6.2. Limitations and Future Research Agenda

The study examined the effects of IGII on WSB, and the function of GKS and GI on the relationship between IGII and WSB. The results enrich the theoretical knowledge of workers’ safety behavior causation. However, the research reported in this paper does have some limitations. Firstly, as IGII is a complex concept and might involve second-order dimensions, this paper viewed the concept as a whole, and further studies could focus on the investigation of its dimension structures. Secondly, the mechanisms by which IGII affects WSB may be more complex, and this study included the effects of GKS and GI, and follow-up researchers could further examine the effects of other concepts on the process, e.g., demographic characteristicsx and safety management commitment.

## Figures and Tables

**Figure 1 ijerph-19-10048-f001:**
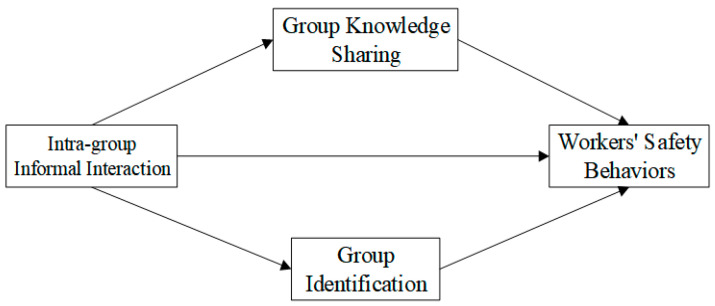
Conceptual model of this study.

**Figure 2 ijerph-19-10048-f002:**
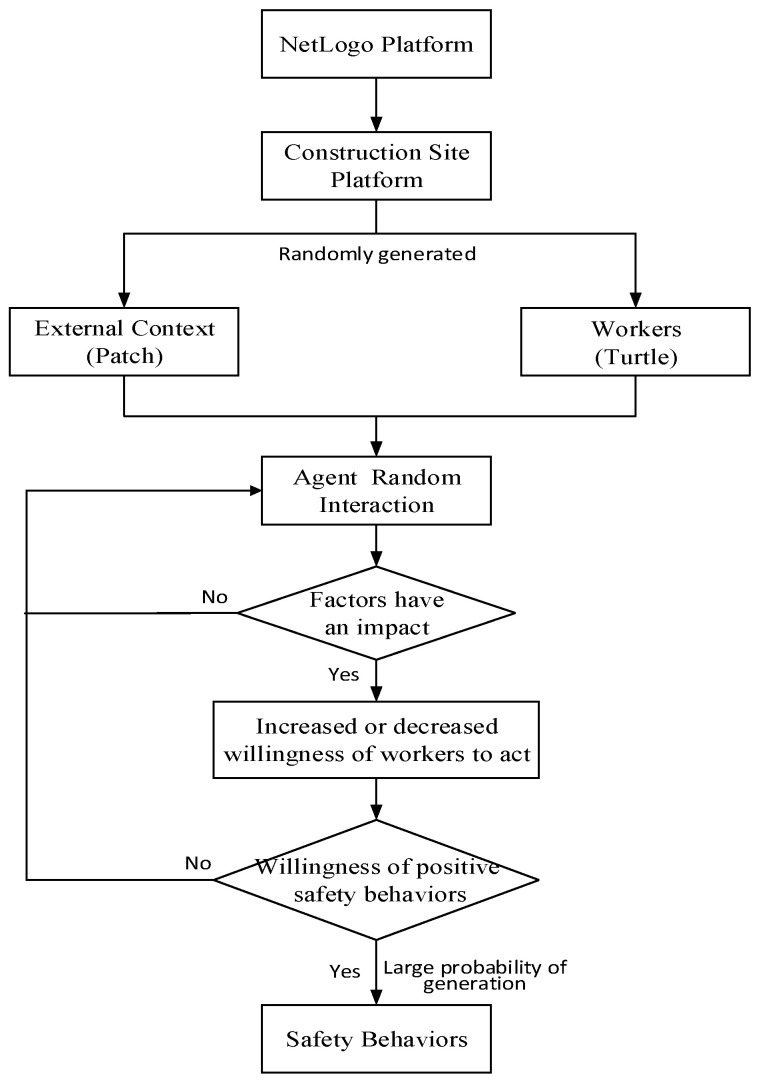
Simulation flow.

**Figure 3 ijerph-19-10048-f003:**
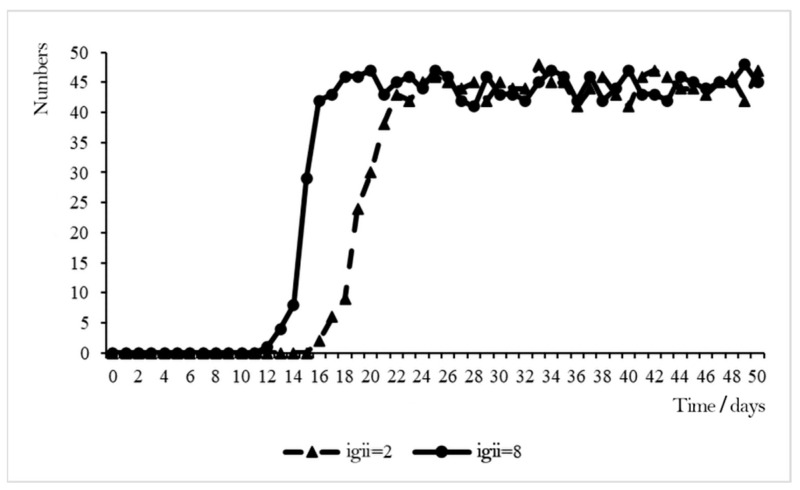
Effect of IGII on WSB.

**Figure 4 ijerph-19-10048-f004:**
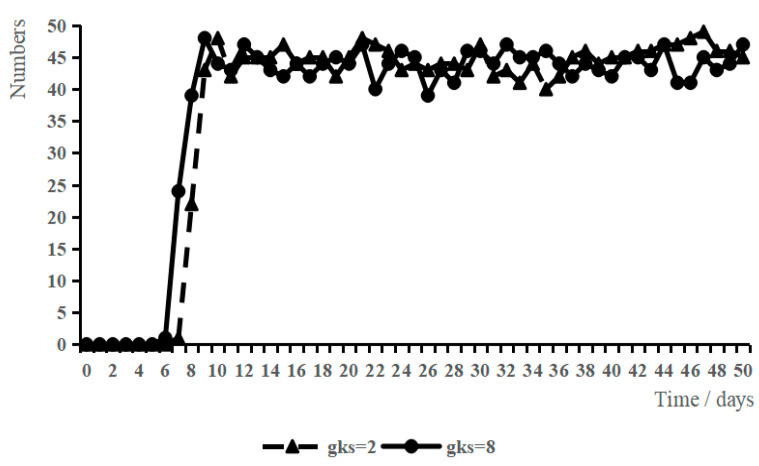
Effect of GKS on WSB.

**Figure 5 ijerph-19-10048-f005:**
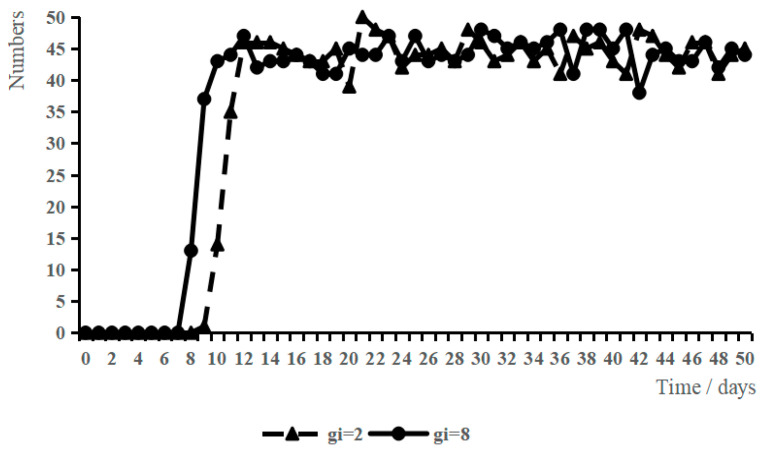
Effect of GI on WSB.4.2. Multi-Variable Experiment.

**Figure 6 ijerph-19-10048-f006:**
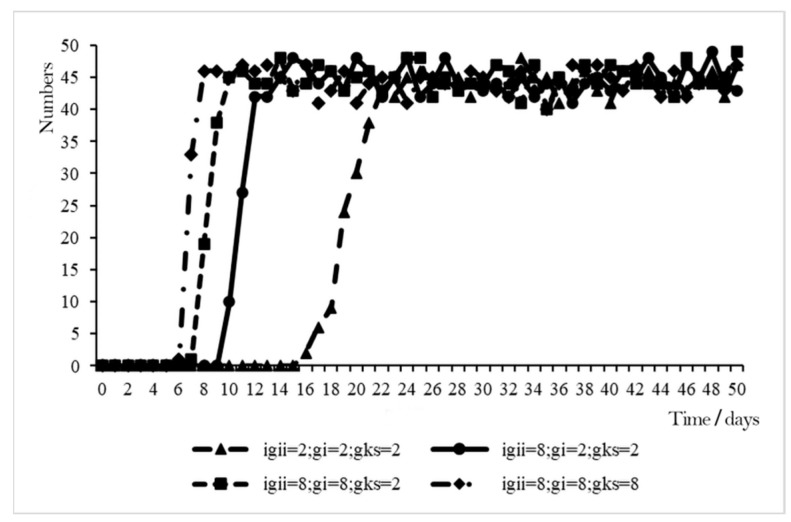
Combined effect of IGII, GI, and GKS on the model.

**Table 1 ijerph-19-10048-t001:** Variable parameter settings.

**Parameter Setting for External Context Variables**					
Variable Name	Variable Explanation	Programming code	Set value	Initial Value	Adjustable or not
Initial IGII	The average level of IGII for the whole construction site; the larger the value, the better the IGII is characterized	initial_igii	0–10	2	Directly adjustable
Initial GKS	The average level of GKS in the whole construction site; the larger the value, the higher the degree of GKS	initial_gks	0–10	2	Directly adjustable
Initial GI	Average level of GI for the whole construction site	initial_gi	0–10	2	Directly adjustable
Patches IGII Properties		patches_igii	N (initial_igii, 1)		Indirectly adjustable
Patches GKS Properties		patches_gks	N (initial_gks, 1)		Indirectly adjustable
Patches GI Properties		patches_gi	N (initial_gi, 1)		Indirectly adjustable
**Agent-related variable parameter setting**					
Variable Name	Variable Explanation	Programming code	Set value	Initial Value	Adjustable or not
Initial number of workers	Number of workers at the construction site at the start of the model run	initial_wokers_count	0–100	50	Directly adjustable
Initial safety behavior costs	Average WSB costs across the construction site, with higher values indicating more effort, resources, and time required for workers to adopt safety behaviors	initial_sb_cost	0–100	45	Directly adjustable
Willingness to initially act safely due to other factors	Other social, organizational, and individual factors at the construction site lead to a willingness to behave safely, with larger values indicating a greater influence	initial_sb_intention	0–100	20	Directly adjustable
WSBC	Individual workers’ safety behavior cost on site	workers_sb_cost	N (initial_sb_cost, 1)		Indirectly adjustable
WSB willingness due to other factors	Willingness to act safely due to other factors of individual workers on site	workers_sb_intention	N(initial_sb_intention, 1)		Indirectly adjustable
Safety behavior generation coefficient	Likelihood of workers adopting safety behaviors when they show a positive willingness to do so	sb_index	B (n, 0.8)		Non-adjustable

## Data Availability

The raw data supporting the conclusions of this article will be made available by the authors, without undue reservation.

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
