# Peer review of "A Simulation Study on the Processes of Intra-Group Informal Interaction Affecting Workers’ Safety Behaviors"

_ijerph, 2022, doi:10.3390/ijerph191610048_

Round 1

Reviewer 1 Report

General comments:

The manuscript "A Simulation Study on the Processes of Intra-group Informal Interaction Affecting Workers' Safety Behaviors" reported the influence of informal interactions in the workgroups on workers' safety behaviors. The author constructed a conceptual model of intra-group informal interaction (IGII), group knowledge sharing, and group identification influencing construction workers' safety behaviors, and then conducted simulation analysis by using the software of NetLogo.

The author stated that intra-group informal interaction can positively influence workers' safety behaviors, both group knowledge sharing and group identification positively influenced the process of IGII affecting safety behaviors. However, the big drawback of this study was the lack of model validation processes. The effectiveness of this model could not be evaluated. The author must quantitatively analyze relevant real-life data to validate the model performance. Thus, I don't think this manuscript is scientific enough for publishing.

Specific comments:

1. Abstract: Line 18: The abbreviations should be used. Line 20-22: It is hard to understand a long sentence, please use short sentences instead.

 2. Introduction:  Line 52-53: Please rephrase, difficult to read.  Line 75: Tense and diction errors, please correct.

 3. Literature Review and Conceptual Model:

(1) Line 117-119: The examples quoted here do not illustrate the above, please rephrase.

(2) Line 142-171: The author explains a lot about WSB in section 2.4, what does WSB specifically mean in this paper? Furthermore, what does WSB include?

(3) Line 172: Please use abbreviation.

(4) Line 181-182: The examples quoted here do not illustrate the above, please rephrase.

(5) Line 185-187: The author stated that IGII can be derived from the formal interaction in the business management area. Please justify the above and explain Significance of this paper.

(6) Line 204: The author stated that IGII can influence GKS, and GKS can influence WSB.

Contradictory to the line 66-67, please rephrase.

(7) Line 214-217: Please explain specifically why these specific actions affect WSB.

(8) Line 222-223: Contradictory to the line 71-72, please rephrase.

(9) Fig1: Please explain the meaning of H1, H2 and H3.

 4. Simulation Experiment Design:

(1) Line 244: Please explain external contextual factors.

(2) Section 3.1:  The specific steps of the simulation should be given.

 5. Simulation experiments:

(1) Section 4.1: According to the simulation results in section 4.1, the SB_COST has a more significant impact on WSB compared to the IGII.

(2) Section 4.1: Please supplement simulation experiment in different GKS.

(3) Section 4.1: Please supplement simulation experiment in different GI.

(4) Fig3, Fig4: Please use English and specify the unit.

(5) Section 4.2: Please supplement and analyse simulation experiment in different safety behavior cost.

 6. Discussion:

(1) Section 5.2: According to the simulation results, it is not possible to state whether GKS or GI has not a direct effect on WSB. Please justify.

(2) Section 5.3: Based on the above simulations, recommendation 3 could not be obtained.

 7. References: References are redundant.

Reviewer 2 Report

The paper provides a model for the worker/supervisor interactions in the construction sector, and how these can be manipulated to improve safety outcomes.  The mathematical model is interesting, and the insights into the Chinese construction also are useful.

I would suggest that one of the premises of the discussion (lines 41 - 54) is flawed.  The western construction sector is operated as a large number of small independent companies.  These companies or "independent contractors" have between 1 and 5 employees.  The senior person is the owner, or tradesperson, or foreman.  A construction site will have many such small companies working on it - each being independent.  This structural nature of the industry and the construction trades makes changing safety behaviours difficult.  I draw your attention to:

https://www.hse.gov.uk/construction/index.htm

https://www.osha.gov/construction

Much of the safety support work has been focussed on changing the behaviours at the individual company level - and the factors that were identified in this paper are those that have been found to work in the west.  As such, I would argue that the paper demonstrates the universal problem in the trades-based construction sector - and that the Chinese construction suffers from the same issues.  

My second concern is that there is little discussion of the importance of the "foreman" in setting the tone for safe behaviours.  The foreman is key - get him to change his behaviours, and his workers will follow.  In the west, we don't perceive this as a "worker choice", but as a failure in "leadership".

I would suggest that reworking the discussion to incorporate more western references, and making a clear demonstration of the importance of leadership will strengthen the paper, and make it more broadly applicable.

Round 2

Reviewer 1 Report

The revised manuscript improved a lot, and I agree to publish in this version.

Reviewer 2 Report

I see that the rewrite of the paper is much improved.  Further, I see the references to the role of the foreman and leadership in setting the tone for safety culture.  These are solid improvements.

I continue to argue that much of the western construction sector is similar to that described in China.  There are small businesses, with the employees being friends or relations of the owner/foreman.  Further, there is a significant section of the construction that uses "undocumented" or "illegal" workers, who usually are relations (or friend of relations) of the owner/foreman.  I understand that the authors have no knowledge of this - and that this information is difficult to quantify and reference, in a typical academic fashion.  

I would suggest that the authors work to find a partner in the west to try to bring together the two worlds - there is an opportunity for a fruitful sharing of information.